# Self- and Cross-Fusing of Furan-Based Polyurea Gels Dynamically Cross-Linked with Maleimides

**DOI:** 10.3390/polym15020341

**Published:** 2023-01-09

**Authors:** Takuya Kumakura, Kenji Takada, Tatsuo Kaneko

**Affiliations:** Graduate School of Advanced Science and Technology, Sustainable Innovation Research Area, Japan Advanced Institute of Science and Technology, 1-1 Asahidai, Nomi, Ishikawa 923-1292, Japan

**Keywords:** bio-based polymers, polyurea, self-healing, furan, dynamic gels

## Abstract

Bio-based polyureas (PUs) with main-chain furan rings were synthesized by the polyaddition of 2,5-bis(aminomethyl)furan with various diisocyanates, such as methylene diphenyl diisocyanate. Several PU’s were soluble in polar organic solvents, and were cast to form thermomechanically stable films with softening temperatures of over 100 °C. The furan rings of the PU main chains underwent a dynamic Diels-Alder (DA) reaction with bismaleimide (BMI) cross-linkers. While the mixed solution of PU and BMI did not show any apparent signs of reaction at room temperature, the DA reaction proceeded to form gels upon heating to 60 °C, which became a solution again by further heating to 80 °C (retro-DA reaction). The solution phase was maintained by rapid quenching from 80 °C to room temperature, while the gel was reformed upon slow cooling. The recovered gels exhibited self-healing properties. A scratch made by a hot knife at temperatures above 80 °C disappeared spontaneously. When two different gels were cut using a knife at room temperature, placed in contact with each other, and heated to 60 °C, they fused. The ability to control the DA/retro-DA reaction allowed gels of varying composition to heal.

## 1. Introduction

Polymer gels have a three-dimensional (3D) network structure that contains a solvent. Owing to their excellent solvent absorption properties, polymer gels have applications in a variety of fields, including biomedical engineering and electronics [1,2,3,4,5,6,7,8]. Conventional polymer gels composed of covalent 3D networks exhibit thermomechanical stability superior to physical gels. However, because the 3D network structure of covalently cross-linked polymers (CLP) is permanent, they lack the ability to undergo reprocessing. One promising approach to overcome this problem is to use dynamic covalent bonds (DCBs) in CLPs [9,10,11]. The introduction of DCBs regenerates covalent bonds inside and at the interface of polymeric materials via transesterification [12], olefin metathesis [13,14], reversible boronate ester formation [15,16], and dichalcogenide bond exchange [17,18]. Furthermore, studies on CLPs containing DCBs have revealed that they exhibit properties such as self-healing [19,20], shape memory [21], and the ability to undergo reversible sol-gel reactions [22]. Self-healing based on bond exchange reactions prevents catastrophic structural collapse and extends the lifetime of materials. Therefore, self-healing polymers have been extensively studied, from their microscopic repair mechanisms to their macroscopic properties [23,24,25,26,27]. Recently, direct adhesion between hydrogels using the inclusion function of cyclodextrins [28] and by covalent bond formation via the Suzuki/Miyaura cross-coupling reaction have been reported [29,30]. Both are based on reactions between functional groups of the side chains. Depending on their composition, CLPs exhibit a range of properties, from soft gels to hard thermoset materials, and are utilized in diverse applications in our daily lives. However, crosslinked polymers cannot dissolve or melt, making it difficult to mix and integrate different crosslinked polymers. To overcome these challenges, it is necessary to employ heterojunction technology for CLPs, where molecular chains are connected in a 3D mesh-like pattern.

In this context, we focused on 2,5-bis(aminomethyl)furan (AMF), a diamine monomer synthesized via catalytic processes from microorganism-derived 5-(hydroxymethyl)furfural (HMF) [31]. Polymers based on AMF introduce a furan in the main chain that provides a site for further functionalization. The furan-maleimide Diels-Alder (DA) reaction allows for reversible dynamic covalent cross-linking to be introduced, which imparts abilities to heal, recycle, and repair the polymeric materials [32,33,34,35,36]. We believe this approach is worthy of investigation, as there are few examples of polymer synthesis focused on chemical modifications of furandiamines and their main-chain furan rings. Unlike furan dicarboxylic acids, we expect AMFs to lend flexibility to the resulting polymers owing to the presence of methylenes adjacent to the furan rings. This may allow for the synthesis of polymers with excellent thermal and mechanical properties. Herein, we focus on a polyurea (PU) formed by the reaction of a diisocyanate and diamine [37,38]. Molecularly, it is similar to polyamide, but has a structure where significant hydrogen bonding occurs between the polymer chains, enhancing the mechanical properties [39,40]. Recently, the use of PUs in films, foams, and composites, such as vertical body panels, spray foams, and microcellular foam liquid plastics, has attracted attention in industrial applications [41,42,43]. A further advantage of PUs is that they do not require catalysts or rapid heating to undergo molding [44].

Here, we describe the synthesis and characterization of a bio-based PU from a microorganism-derived AMF via polyaddition reaction with various diisocyanate compounds. Some PUs showed thermally reversible sol-gel transitions, achieved by adding cross-linking agents. In particular, heterojunctions between different PU gels are formed by the dynamic covalent bonding of CLPs while maintaining the gel state by heating. This design includes a furan group in the main chain and cross-linking reactions of different polymer chains by the DA reaction, which are expected to exhibit the high performance and self-healing properties.

## 2. Materials and Methods

### 2.1. Materials

2,5-Bis(aminomethyl)furan (99% purity) was obtained from Nippon Shokubai Co., Ltd. (Tokyo, Japan). Methylene diphenyl diisocyanate (MDI), 2,4-toluene diisocyanate, 2,6-toluene diisocyanate, 1,3-phenylenediisocyanate, 1,4-phenylenediisocyanate, *m*-xylylene diisocyanate, 1,5-diisocyanatonaphthalene, dicyclohexylmethane 4,4′-diisocyanate, isophorone diisocyanate, hexamethylene diisocyanate, and 4,4′-diphenylmethane bismaleimide (BMI) were purchased from Tokyo Chemical Industry Co., Ltd. (Tokyo, Japan) 4,4′-Diisocyanato-3,3′-dimethylbiphenyl and *N,N*-dimethylacetamide (DMAc) were purchased from Kanto Chemical Co., Inc. (Tokyo, Japan), 1,4-Butane diisocyanate and 1,8-octane diisocyanate were purchased from Sigma-Aldrich Co., Inc., Merk (Darmstadt, Germany). *N*-Methyl pyrrolidone (NMP), dimethyl sulfoxide (DMSO), *N*, *N*-dimethylformamide (DMF), tetrahydrofuran (THF), acetone, chloroform (CHCl_3_), dichloromethane (DCM), ethanol (EtOH), *m*-cresol, trifluoroacetic acid (TFA), and concentrated sulfuric acid (conc. SA) were purchased from Wako Pure Chemical Industries Ltd. (Osaka, Japan) All reagents were used without further purification.

### 2.2. Instruments

^1^H NMR measurements were performed using a Bruker Bio-spin AG 400 MHz spectrometer. FT-IR spectra were used with a Perkin–Elmer Spectrum One spectrometer (Waltham, MA, USA), between 4000 and 600 cm^−1^, using a diamond-attenuated total reflection accessory. The gel permeation chromatography (GPC) were performed on a Shodex column (SB 806M × 2), column oven (GL Science, Tokyo, Japan, CO 631A, set at 40 °C), a degassing unit (GL Science, DG 660 B), a pump (JASCO, Tokyo, Japan, PU-2080 Plus), a refractive index detector (JASCO, 830-RI), and an ultraviolet detector (JASCO, UV-2075 plus) using a 0.01 mol L^−1^ LiBr solution in DMF as an eluent (flow rate, 1 mL min^−1^). Thermal gravimetric analysis (TGA) was carried out using a HITACHI STA 7200 under nitrogen flow (flow rate, 200 mL min^−1^) from 25 °C to 800 °C (heating rate, 5 °C min^−1^) to determine the 5% and 10% decomposition temperatures (*T*_d5_ and *T*_d10_). Differential scanning calorimetry (DSC) was performed with a SEIKO X-DSC7000T, SII Seiko Instruments. Inc. (Chiba, Japan) to measure the glass transition temperature (*T*_g_), from 25 to 300 °C at a heating rate of 10 °C min^−1^ with ca. 5 mg of sample. The ultraviolet-visible (UV-Vis) absorption spectroscopy was performed with a V-670 (JASCO) instrument in the 200–800 nm range. Finally, a tensile testing machine (INSTRON 3365-L5) was performed with a 1 mm s^−1^ movement at room temperature. The rectangular polymer film dimensions were length 20 mm, width 5.0 mm with a thickness of 2.5 μm.

### 2.3. Syntheses of Bio-Based PUs Using AMF

For synthesis of PUs, 2,5-bis(aminomethyl)furan (AMF) (126.2 mL, 1.0 mmol) was dissolved in dimethylacetamide (DMAc, 2.0 mL) and dropped, very slowly, into an MDI (250.3 mg, 1.0 mmol) solution in DMAc (2.0 mL) with stirring (Figure 1). An ice–water bath cooled the mixture. The obtained solution had sufficient viscosity approximately 2 h after the drop, which indicated formation of the PUs. The solution was diluted with a DMAc solution before reprecipitation by dropping into methanol. The precipitate was collected by suction filtration and dried under reduced pressure for 12 h. Powdery or fibrous white polyurea resulted (365.1 mg, yield 97.0%, Figure 1). We synthesized PUs from AMF and other diisocyanate monomers using an analogous procedure (Figure 1).

### 2.4. Preparation of PU Gel with BMI

PU-1 (100.0 mg, 264.9 mmol) was dissolved into DMAc (1.5 mL), which was added into a solution of 4,4′-bismaleimidodiphenylmethane (BMI) (9.5 mg, 26.5 mmol) in DMAc (0.5 mL). The reaction was carried out at room temperature before heating to 50 °C with stirring for 1 h. Yellow gels formed as the reaction progressed. When the gel was heated to 100 °C, it disappeared (Figure 2).

## 3. Results and Discussion

### 3.1. Syntheses and Performance of PUs

PUs with main-chain furan rings were prepared by the polyaddition of AMF with a stoichiometric amount of diisocyanate. The weight-average molecular weight (*M_w_*), number-average molecular weight (*M_n_*), and molecular weight distribution (*M_w_*/*M_n_*) were determined using GPC. A summary of the results is in Appendix A. The molecular weights were high enough to confirm successful polyaddition. The solubility of PUs was evaluated using non-polar solvents such as hexane, chloroform, and dichloromethane; polar protic solvents such as distilled water, methanol, and ethanol; and polar aprotic solvents such as acetone, THF, DMF, DMAc, DMSO, and NMP. Compounds PU-1, PU-2, PU-3, PU-4, and PU-5 were soluble in polar aprotic solvents such as DMF, NMP, DMAc, and DMSO at room temperature (Table 1). ^1^H NMR measurements of the solubilized PUs revealed peaks at 6.5 and 8.5 ppm, specific to urea bonding (Appendix A). In addition, FT-IR measurements showed absorption peaks at 1450 cm^−1^ for the C-N stretch, 1670 cm^−1^ for the C=O stretch, and 1620 cm^−1^ and 3450 cm^−1^ for the N-H stretch [45], indicating that the urea bond was formed and the desired polyurea was obtained (Appendix A). PU films were prepared by casting them onto glass plate from a 50 mol L^−1^ DMAc solution and allowed them to dry at 100 °C for at least 1 h (Appendix A). The 1% and 10% weight loss temperatures (*T*_d1_ and *T*_d10_, respectively) were determined by thermogravimetric analysis in a nitrogen atmosphere at a heating rate of 10 °C min^−1^. The *T*_d1_ values of the various PUs were above 240 °C, and their DSC measurements revealed *T*_g_ above 110 °C (Appendix A). The results of the tensile tests of the PU-1, PU-2, PU-3, PU-4, and PU-5 indicated all the films were tough and flexible. In particular, PU-1 exhibited the best mechanical properties, with a tensile strength of 65 MPa and a Young’s modulus of 820 MPa. The thermal and mechanical properties indicate that the PUs prepared here were more thermoresistant and harder than conventional aliphatic PUs. We attribute this to the aromaticity of the furan ring and intermolecular hydrogen bond interactions [46,47].

### 3.2. Dynamic Cross-Linking

BMI was added into PU-1 dissolved in DMAc and allowed to cross-link via the DA reaction. When BMI was mixed with PU-1 and stirred at room temperature, no cross-linking was observed, even after several days. The mixture became a gel upon heating to 50 °C, and its apparent elasticity increased with time. The gelation was observed at 50, 60, 70, and 80 °C, and the gelation time decreased at higher temperatures. However, when heated between 85 and 95 °C, a small amount of solution began to elute from the gel, and above 95 °C, the gel returned to solution in approximately 10 min (Figure 1). This may be due to the decomposition of the cross-linked structure via retro-Diels-Alder (retro-DA) reaction. Subsequently, the solution gelled again by cooling to 80–60 °C over approximately 1 h, indicating a reversible process (Figure 1). The DA reaction was observed by ^1^H NMR measurements of PU-1 (Figure 2a) and a mixture of PU-1 with BMI at various temperatures (Figure 2b–d), showing the reaction of PU with the crosslinkers. In Figure 2a,b, all signals in the spectra obtained at 25 °C to the protons of PU and BMI were assigned. At 60 °C, proton signals were broadened, except for unreacted free BMI, indicating gelation. Additionally, proton signals from cross-linking points were barely detected at 3.6 ppm (m) and 6.4 ppm (l) (Figure 2c). Upon heating to 100 °C, these same signals reappeared clearly, and there was no detection of cross-linking points (Figure 2d), confirming both the retro-Diels-Alder reaction and gel-sol transition. The compressive strength of the PU gels depended on the amount of BMI added, and we obtained gels with high elasticity when adding more than 10 mol % BMI (Appendix A). Among the gels, those containing 80 mol % BMI showed an initial modulus of elasticity of 0.8 kPa. In contrast, the modulus of elasticity tended to decrease from 0.2 to 0.5 kPa when the amount of BMI exceeded 100 mol % (Appendix A). This attribution was an undesired reaction between excess BMI and the furan structure in the main chain, which makes it difficult to form a cross-linked structure.

Additionally, the solution did not gel upon quenching from 80 °C to about 15 °C in a water bath, even after 24 h of incubation. One reason for the hysteretic behavior of gelation is the slow recovery of the gel state, and the molecular mobility is too low to allow for a DA reaction to occur at room temperature. When a substituent is attached to either the 2- or 5-position of the furan ring, its reactivity is very high, whereas when substituents are attached to both positions, the reactivity is reported to be lower than monosubstituted examples [48]. In addition, the furan structure in the main chain has a lower molecular mobility than the substituted furan ring in the side chain. The low reactivity of 2,5-disubstituted furan may be related to the retro-DA reaction under conditions present at 80 °C. A thermoreversible sol-gel transition was also observed in DMAc, DMSO, and NMP. Soluble PUs other than PU-1 (PU-2 to PU-5) showed similar gelation behavior toward DMAc (Appendix A).

### 3.3. Healing Behavior

A knife heated to approximately 100 °C was used to scar the PU-1 gel. The gel then stood at room temperature, and over a period of 1 to 2 min, the surface scar gradually disappeared (Figure 3). Spontaneous healing occurred during cooling, as seen in the photograph at the right side of Figure 3. From this result, it was concluded that the heated knife turned the surface into a solution via retro-DA reaction, which flowed and filled the incision. Healing then occurred due to the dynamic DA reaction.

Next, the formed gels were cut with an unheated knife at approximately 20 °C and the cut surfaces placed in contact with one other at room temperature. Healing did not occur. When the cut surfaces were placed in contact with each other for 1 h at 60 °C, adhesion of the cut surfaces was confirmed. The adhesive was pulled using a spatula; however, the adhesive did not detach from the cut surfaces, and the gel was not torn at the bonded area (Figure 4). This phenomenon is different from healing based on the gelation of the solution, as the sample is already a gel at room temperature. In other words, self-healing occurred entirely within the gel state, where the DA reaction of the network occurred at the gel surface (Figure 4a). The gels PU-1 and PU-3, having different types of network main chains, were cut using a knife at room temperature and placed in contact with each other at 60 °C. Healing was observed, forming a heterojunction between PU-1 and PU-3. The pigments in each gel diffused throughout (Figure 4b). Usually, different polymers are not miscible with each other, as both networks are unable to intercalate at the molecular level. Such cross-healing behavior can only be based on the DA reaction on the outer surface of each gel.

## 4. Conclusions

The synthesis of furan-containing PUs by a polyaddition reaction of AMF with thirteen different diisocyanate compounds were reported. In the dry state, the PUs exhibited greater thermoresistance and hardness than conventional ones. Five PUs were soluble in polar solvents such as DMSO, DMAc, DMF, and NMP. The existence of a reversible DA reaction between the main chain furan ring and maleimide of the crosslinkers to generate a network structure was confirmed by NMR. The PUs were not gelated in the presence of bismaleimide at room temperature, but rather by heating to 50–80 °C. At temperatures higher than 80 °C, the gel transforms into a solution state as a result of retro-DA reactions. The solution showed gelation behavior with slow cooling but maintained the solution state upon rapid quenching to room temperature. Dynamic covalent bonding extended to the healing behavior of the gels. While the cut surfaces did not adhere to each other at room temperature, we observed self-healing behavior at 60 °C. We also confirmed this healing phenomenon when we placed the cut surfaces of different PU gels in contact at 60 °C. This suggests that the outer surface of the cross-linked networks underwent a chemical reaction to form a heterojunction. The application of this phenomenon can be extended to the fabrication of self-healing films, gels, and elastomers.

## Data Availability

The data presented in this study are available on request from the corresponding author.

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
