# Peer review of "Self- and Cross-Fusing of Furan-Based Polyurea Gels Dynamically Cross-Linked with Maleimides"

_polymers, 2023, doi:10.3390/polym15020341_

Round 1

Reviewer 1 Report

Dear Author,

The article titled 'Self- and Cross-fusing of Furan-based Polyurea Gels Dynamically Cross-linked with Maleimides" is an interesting read.

Here are my comments:

(1) How many times stress strain curve taken (Supporting information). Stress strain are not smooth, so wondering how you calculate compression modulus and other properties. 

(2) The author loosely states in their analysis that it's reversible diels-alder reaction and let the reader to find the reversible process. I strongly suggest to explain in detail NMR spectra and show how this reversible reaction is happening.

(3) This type of sol-gel type of Diels-alder has been done before, can author describe what is new in their findings.

(4) In line 169, what does quenting means? 

Author Response

Dear Associate Editor of Polymers,

Assistant Editor Ms. Lisbeth Wang,

We greatly appreciate the helpful comments from reviewers related to our manuscript entitled “Self- and Cross-fusing of Furan-based Polyurea Gels Dynamically Cross-linked with Maleimides (manuscript ID: polymers-2103523; For “State-of-the-Art Polymer Science and Technology in Japan (2021,2022)”)”.

We carefully revised the manuscript according to reviewer’s suggestions and comments. In addition, we revised some erratum in main text and English collection was carried out by Editage. In the cover letter, the page and line of the article was referred to the revised version. The responses to the reviewers were attached from the next page. We now hope that you will accept the revised version for publication in Polymers.

Finally, these reviewer’s comments and suggestions were very helpful. The revised manuscript is greatly improved from the previous one. We hope that you will consider it acceptable for publication in Polymers.

Thank you once again for your kindness and generosity with your time.

Sincerely yours,

Tatsuo Kaneko, Ph. D.

Professor

Japan Advanced Institute of Science and Technology

Japan

Reviewer 2 Report

Kaneko et al are reporting biobased polyureas (PU) with furan rings in the main chain were synthesized by a poly- 8 addition of 2,5-bis(aminomethyl)furan with various diisocyanates such as methylene diphenyl 9 diisocyanate. This is an interisting work on smart gels with tunable properties. Authors should improve these parts:

1. 1H-NMR caracterization, highlight the peaks by assigning the corresponding molecules, dissappearance and appearance of new peaks for the figure 2

2. Rheological characterizations are missing, these are very important when gels are concerned.

3. The self-healing should be supported by a proof of mechanical or viscoelastic property recovery.

Author Response

(The authors gave the same response as above.)

Round 2

Reviewer 1 Report

Dear Author,

 The author has incorporated and explained in detail the required explanation for the manuscript as well as for readers. 

This article is now in a form to be accepted.

Reviewer 2 Report

Dear Editor,

The manuscript entitled Self- and Cross-fusing of Furan-based Polyurea Gels Dynamically Cross-linked with Maleimides by Takuya Kumakura , Kenji Takada , Tatsuo Kaneko can be published in the current version as the requested clarifications/ inputs were adressed.